# Short-Term Sources of Cross-Linguistic Phonetic Influence: Examining the Role of Linguistic Environment

**Daniel J. Olson** 

School of Languages and Cultures, Purdue University, West Lafayette, IN 47907, USA; danielolson@purdue.edu

**Abstract:** While previous research has shown that bilinguals are able to effectively maintain two sets of phonetic norms, these two phonetic systems experience varying degrees of cross-linguistic influence, driven by both long-term (e.g., proficiency, immersion) and short-term (e.g., bilingual language contexts, code-switching, sociolinguistic) factors. This study examines the potential for linguistic environment, or the language norms of the broader community in which an interaction takes place, to serve as a source of short-term cross-linguistic phonetic influence. To investigate the role of linguistic environment, late bilinguals (L1 English—L2 Spanish) produced Spanish utterances in two sessions that differed in their linguistic environments: an English-dominant linguistic environment (Indiana, USA) and a Spanish-dominant linguistic environment (Madrid, Spain). Productions were analyzed at the fine-grained acoustic level, through an acoustic analysis of voice onset time, as well as more holistically through native speaker global accent ratings. Results showed that linguistic environment did not significantly impact either measure of phonetic production, regardless of a speaker's second language proficiency. These results, in conjunction with previous results on long- and short-term sources of phonetic influence, suggest a possible primacy of the immediate context of an interaction, rather than broader community norms, in determining language mode and cross-linguistic influence.

**Keywords:** bilingualism; phonetics; language mode; cross-linguistic influence; transfer; voice onset time; global accent rating

## 1. Introduction

Research has shown that bilinguals, including both early bilinguals (e.g., MacLeod and Stoel-Gammon 2005) and late second language learners (e.g., Schmid et al. 2014), can effectively maintain two separate phonetic systems for their two languages. However, these two phonetic systems are not fully independent, and cross-linguistic influence, in which the phonetic system of one language is influenced by the competing language, has been evidenced across a range of bilingual populations and contexts. Importantly, there are a variety of both long-term and short-term sources of cross-linguistic influence (i.e., transfer), impacting both a bilingual's first (L1) and second (L2) languages. Broadly, short-term refers to contexts in which production or perception may be altered for a single speaker in response to immediate or momentary changes in the linguistic situation (e.g., bilingual language mode and code-switching), while long-term refers to sustained influences over longer periods of times (e.g., acquisition and immersion). While some long-term sources of phonetic cross-linguistic influence, such as immersion (e.g., Casillas 2020) and instruction (e.g., Lee et al. 2015), are well-studied, less research has focused on potential short-term (i.e., transient) sources of cross-linguistic influence (Simonet 2014).

Given the previous focus on long-term sources of cross-linguistic phonetic influence, and the emerging research showing the relevance of a number of short-term sources, the current study examines the potential impact of a novel source of such influence: linguistic environment. Linguistic environment is broadly defined as the language norms of the broader community in which an interaction or experimental paradigm is conducted. Bilinguals naturally move from one linguistic environment to another for work, travel, and social interaction, and such shifts in context or environment may serve to foster cross-linguistic influence at the phonetic level. This study adds to our theoretical understanding of the organization of bilingual phonetic systems, and highlights both the sources of and limits on cross-linguistic phonetic influence.

## 2. Literature Review

Cross-linguistic influence at the phonetic level can be described as the way in which the phonetic system of one of a bilingual's languages impacts the production and perception of speech sounds in their other language (Jarvis and Pavlenko 2008). Within studies of bilingual phonetics, and as a condition for examining cross-linguistic influence or transfer, early research sought to establish that bilinguals are indeed able to produce and maintain separate phonetic norms in each of their two languages (e.g., Caramazza et al. 1973). While bilinguals produce different phonetic categories for their two languages, the relationship of these categories to the monolingual norms may depend on a variety of factors. For example, some research has shown that bilinguals, particularly early bilinguals, may show little to no deviance from the monolingual targets in each of their languages (e.g., Flege et al. 1999; Guion et al. 2004; Piske et al. 2002), while others have found that late bilinguals (e.g., Flege and Eefting 1987; Flege and Port 1981) and even some early bilinguals (e.g., Flege et al. 1995; Fowler et al. 2008) produce phonetic categories that deviate from those produced by monolingual speakers.

### 2.1. Long-Term Sources of Cross-Linguistic Phonetic Influence

In the line of research that has examined cross-linguistic phonetic influence, there has been a significant body of research that has examined the impact of L1 phonetic systems on L2 phonetic categories. Broadly, this line of research has established that the extant L1 system exerts influence over the L2 system, shaping both production and perception of L2 phonetics. Several L2 phonological models provide theoretical accounts for the mechanisms that govern the acquisition of new phonetic categories, and there is broad agreement that the ability to acquire a new L2 category depends on the relationship to existing L1 sounds (e.g., Speech Learning Model (Flege 1987, 1988, 1991, 1995); Native Language Magnet theory (Kuhl 1992, 1993a, 1993b); Perceptual Assimilation Model-L2 (Best and Tyler 2007)). Moreover, as the L2 phonetic system develops as the result of engagement with the L2, either through immersion (for review of study abroad and at-home instruction, see Casillas (2020); for longer-term immigration, see Piske et al. (2001); for study abroad, see Solon and Long (2018))[1] or instructed acquisition (for review of L2 phonetic instruction, see Lee et al. (2015)), the cross-linguistic influence of the L1 on the L2 diminishes and the L2 becomes more native-like. As shown in research on study abroad, such effects may be observed following stays of just a few weeks (e.g., Lord 2010), although Solon and Long (2018) note significant individual variation. Following long-term exposure, ultimate phonetic attainment may (e.g., Schmid et al. 2014) or may not (e.g., Flege 1987, 1991) match native speaker norms (for variability in attainment, see Simon (2009)).

Beyond the expected influence of long-term exposure to the L2 on L2 phonetic production, there is also evidence of influence of the L2 on the L1 phonetic system. Several studies have found an

---

[1] For a discussion on the interaction between length of residence and other variables, notably age of acquisition, see Piske et al. (2001). As Piske et al. (2001) note, length of residence "only provides a rough index of overall L2 experience" (p. 197). While many studies talk about immersion, immigration, and length of residence, these may be used as a proxy for overall L2 experience. For the current study, it is relevant that these factors, whether conceptualized as L2 experience or length of residence, function as long-term sources of linguistic change and interaction.

inverse relationship between the length of residence in an L2 linguistic environment or L2 proficiency and the degree of phonetic influence of the L2 on the L1 (e.g., Bergmann et al. 2016; Major 1992; Stoehr et al. 2017). Speakers that had spent longer in the L2 linguistic environment, or those with greater proficiency in the L2, evidenced greater degrees of L2 phonetic transfer to the L1 (e.g., Major 1992), leading Major (1992) to claim that the L1 phonetic system "is not a fixed and stable system but rather a fluid and changeable one that is highly subject to the influence of a well-developed second system" (Major 1992, p. 204). Other research has shown a degree of bidirectional cross-linguistic influence (e.g., Fowler et al. 2008), in which the L1 influences the L2 and the L2 influences the L1. Again, cases of L2 to L1 transfer have most often been found following long-term engagement with the L2.

Although much of this research has focused on a linear trajectory, such as tracking the shift in L2 production towards monolingual norms over time, it is clear that such long-term shifts are dynamic. In their seminal work, Sancier and Fowler (1997) tracked the phonetic production of a single Portuguese-English bilingual speaker as they moved between multiple linguistic environments. Their results showed that, following a stay of several months in a Portuguese linguistic environment, voice onset times (VOTs) for both Portuguese and English productions became shorter and more Portuguese-like, a phenomenon referred to as "gestural drift" (Sancier and Fowler 1997, p. 422). In contrast, following a stay of several months in an English linguistic environment, VOTs for both languages became longer and more English-like. These results demonstrate that a long-term change in linguistic environment can promote a degree of phonetic interaction, whereby both of the bilingual's languages are impacted by the language of the ambient environment. In short, cross-linguistic phonetic influence, as evidenced by bilingual gestural drift, appears to be dynamic and responds long-term to factors in the ambient linguistic environment.

The existing studies on long-term immersion, study abroad, and gestural drift generally consider changes in L1 and L2 phonetic production following a significant length of stay in the L2 linguistic environment, from weeks or months (e.g., Díaz-Campos 2004; Lord 2010; Nagle et al. 2016) to years (Piske et al. 2001). As noted by Simonet (2014), "The vast majority of work on interlingual or cross-linguistic phonetic influence in bilingualism does not explicitly distinguish between long-term and transient interference. Albeit implicitly, most studies explore features that are attributed to long-term interference" (p. 27). Yet more recent research has shown that the degree of cross-linguistic phonetic influence may also be subject to short-term variables.

## 2.2. Short-Term Sources of Cross-Linguistic Phonetic Influence

While the previous research detailed above highlights the notion that phonetic influence between a bilingual's two languages may occur as the result long-term factors, more recent research has begun to examine short-term sources of cross-linguistic phonetic influence. As Simonet (2014) notes, other authors use somewhat different terminology to refer to the same or similar phenomenon (p. 27). Grosjean (2012) differentiates between static and transient sources of cross-linguistic influence, while Paradis (1993) differentiates between competence and performance related cross-linguistic interference. For reasons noted in Simonet (2014), the long-term vs. short-term distinction is used henceforth. Here, *short-term* is used in contrast to long-term, broadly referring to situations or contexts in which phonetic production or perception is altered for a single speaker in response to immediate changes in the broader linguistic situation. These short-term factors may include a shift in the languages used in an interaction, also called the language context (e.g., Olson 2016), changes in external sociolinguistic factors, and even use of cognate tokens.

Several studies have begun to examine the potential of the language of a given interaction or experimental session as a source of short-term cross-linguistic phonetic influence. Simonet (2014), for example, examined the production of Catalan vowels by Spanish-Catalan bilinguals across two different sessions: unilingual and bilingual. In the unilingual session, all stimuli—utterances containing the target Catalan tokens—were drawn from Catalan, as were the session instructions. In contrast, stimuli in the bilingual session included utterances containing the target words from both Spanish and

Catalan, in random order. It is important to note that the target tokens were embedded in meaningful utterances, and switches between Catalan and Spanish took place only at the utterance level. As such, the tokens under consideration were all non-switched tokens. Results from the production experiment revealed that Catalan vowels differed between the monolingual and bilingual sessions, with Catalan vowels becoming more Spanish-like in the bilingual session. Similarly, in a cued picture-naming paradigm, Olson (2013) compared the VOT of English and Spanish tokens produced in a monolingual session (i.e., 95% English—5% Spanish or 95% Spanish—5% English) with tokens produced in a bilingual session (i.e., 50% English—50% Spanish). Results showed differences in phonetic production of VOT depending on the nature of the session. Non-switched productions in the bilingual contexts, specifically in a participant's L1, shifted in the direction of the opposite language, with English VOTs becoming more Spanish-like and Spanish VOTs becoming more English-like. A similar effect may be seen in perception, in which the auditory context (i.e., English-like or Spanish-like acoustic features) surrounding a given ambiguous token serves to engage a language-specific perceptual system (Gonzales and Lotto 2013; for perceptual boundaries and language modes, see Casillas and Simonet (2018)). Further research has shown that while bilingual experimental sessions may foster cross-linguistic phonetic influence, the nature of this transfer may be dependent on a speaker's proficiency or dominance (Amengual 2018). Notably, Amengual (2018) found that bilinguals were more likely to experience cross-linguistic transfer resulting from a bilingual paradigm in their less dominant language.

These results from experimental paradigms also find some preliminary support from a more naturalistic paradigm. In her study of bilingual English-Arabic speaking children, Khattab (2002) collected naturalistic data in both English and Arabic-oriented language sessions. The results showed that while children clearly differentiated between the two phonetic systems, particularly with respect to /r/, English tokens produced during the Arabic sessions underwent a degree of phonetic transfer, becoming decidedly more Arabic-like. Subsequent analysis suggests that such transfer may relate to the language dominance of the interlocutor (Khattab 2009), with Arabic-accented English used during interactions with Arabic-dominant listeners. These results suggest that the use of two languages in the same interaction may serve to promote a degree of cross-linguistic influence.

Further evidence for cross-linguistic phonetic influence arising from the use of two languages in a single interaction can be seen in work on the phonetics of code-switching. *Code-switching* refers to the alternation between two or more languages or language varieties in a single discourse (e.g., Myers-Scotton 1997). As such, code-switching represents a clear point in an interaction in which both languages are simultaneously (or nearly simultaneously) activated and serves as a potential short-term source of cross-linguistic phonetic influence. Unlike the bilingual sessions detailed above (e.g., Khattab 2002; Olson 2013; Simonet 2014), which varied the ratio of the language used in a given session or block but examined non-switched productions, research focused on the phonetics of code-switching has generally focused on the potential for phonetic transfer at or near the point of switch. A growing body of research has begun to establish that code-switching impacts phonetic production at the segmental level, most notably inducing a degree of phonetic transfer (although, for a lack of transfer, see Grosjean and Miller (1994) and Muldner et al. (2019)). The exact nature of the phonetic influence found has varied. Several studies have found evidence of unidirectional transfer at the point of switch (Antoniou et al. 2011; Balukas and Koops 2015; Bullock et al. 2006), with Language A shifting in the direction of Language B, but Language B failing to show evidence of transfer. Other studies have found bidirectional transfer (Bullock and Toribio 2009; González-López 2012; Olson 2016; Schwartz et al. 2015), with Language A shifting in the direction of Language B and Language B shifting in the direction of Language A (for an account of unidirectional vs. bidirectional transfer, see Olson (2019)). This cross-linguistic phonetic influence has been found across a variety of paradigms, including naturalistic (e.g., Balukas and Koops 2015) and read speech (e.g., Antoniou et al. 2011), and for different types of code-switches (e.g., for single-word insertions, see Olson (2016); for alternational code-switching, see Bullock and Toribio (2009)). These shifts are

largely phonetic in nature, rather than phonological, and bilinguals generally do not implement phonological categories of the opposite language. Thus, code-switching, which activates both of a bilingual's languages in a compressed timeframe, appears to serve as a short-term source for bilingual phonetic influence.

An additional case for short-term phonetic influence driven by linguistic factors can be seen in the production of cross-linguistic cognates. Cognates, words that have significant cross-linguistic overlap in meaning, phonology, and orthography (Amengual 2018), may result in the activation of both language systems, and as such represent a possible short-term source of cross-linguistic influence. Cognates have been shown to be produced with a degree of phonetic transfer. Amengual (2012), for example, showed that Spanish-English bilinguals produced longer (i.e., more English-like) VOTs in Spanish for cognate words than non-cognate words, a finding that held for heritage speakers (i.e., early bilinguals) and both late L1 English—L2 Spanish and L1 Spanish—L2 English bilinguals (for Spanish-Catalan bilinguals, see Amengual (2016)). Again, as cognates may activate both languages, they can be seen as a short-term source of cross-linguistic phonetic influence.

Changes in the language of the paradigm or interaction (e.g., Simonet 2014), code-switching (e.g., Olson 2016), and cognate status represent cases in which a short-term or immediate shift in the linguistic content of an interaction favors a degree of cross-linguistic phonetic influence. Yet, there is some evidence that non-linguistic changes in the external environment may also impact linguistic behavior. Hay and Drager (2010), for example, found that including region-specific objects in the experimental environment, such as a stuffed kangaroo (i.e., Australia-specific) or kiwi (New Zealand-specific), impacted vowel perception. The authors suggest that objects in the "ambient environment" can impact participant phonetic perception (p. 889). Other studies have found that visually salient characteristics of a speaker may influence phonetic perception (e.g., for intelligibility and accentedness, see Babel and Russell (2015)). For example, in a perceptual experiment, Koops et al. (2008) found that listeners' phonetic perception of stimuli reflecting an on-going, age-graded phonetic change (i.e., PIN-PEN unmerger in Houston, TX, USA) depended on perceived speaker age. Paralleling the change in progress in the local community, listeners were more likely to assume merged phonetic categories for older speakers and unmerged categories for younger speakers. The impact of such external factors on phonetic perception also extends to non-visible social information, such as supposed geographic origin (e.g., Niedzielski 1999). In these cases, it is not the linguistic content of an interaction or paradigm that shifts, but rather the surrounding environment and/or perceived interlocutor.

It is worth considering these short-term sources of cross-linguistic influence within the framework of a bilingual's language modes (e.g., Grosjean 1998, 2001, 2008). Bilinguals have the ability to operate along a linguistic continuum from operation entirely in Language A (i.e., monolingual mode) to operation in Language B (i.e., monolingual mode), including a variety of bilingual modes in which each of the two languages may be used to differing degrees. Language mode has been described in terms of the relative activation of each of the bilinguals two languages, with monolingual mode involving the activation of only (or predominantly) one language and a balanced bilingual mode involving the roughly equal activation of the two languages (Grosjean 2008). Grosjean (2008) notes that a speaker's language mode may be impacted by a variety of factors, including the "form and content of the message," the language act, the interlocutors, and the general situation of the interaction. All of these factors may be considered "short-term," in that they are subject to change from day-to-day or even interaction-to-interaction for a single bilingual speaker. Moreover, shifts in a speaker's (or listener's) position on the language mode continuum may impact their language production (or perception) patterns (e.g., Soares and Grosjean 1984). The previous findings of differing levels of cross-linguistic phonetic influence presented above can be reconceptualized as resulting from differing language modes. Short-term sources of cross-linguistic influence can be seen as variables that cause an immediate shift in a bilingual's position along the language mode continuum. Thus, short-term variables, including the language(s) of a given interaction or paradigm, the use of code-switching, and changes in the surrounding environment, may effectively serve to manipulate the relative activation (or suppression)

of a bilingual's two languages, with more equal activation resulting in greater degrees of cross-linguistic phonetic influence.

## 2.3. Research Questions

Previous research has established that bilinguals, including late L2 learners, establish different phonetic norms for their two languages. Moreover, there appear to be both long-term (e.g., acquisition and immersion) and short-term (e.g., bilingual mode, code-switching) sources of phonetic cross-linguistic interaction. While long-term immersion in a given linguistic environment has been shown to impact the degree of cross-linguistic influence, the current study examines the potential for linguistic environment to act as a short-term source of cross-linguistic phonetic influence. That is, whether a change in linguistic environment results in an immediate shift in the degree of cross-linguistic phonetic influence. Two specific research questions are addressed:

RQ1: Does a short-term change in the linguistic environment impact phonetic production? For this study, a short-term change in linguistic environment is operationalized as a single speaker moving from an English-dominant linguistic environment to a Spanish-dominant linguistic environment (or vice-versa).

**Hypothesis 1.** *Drawing on previous research that has shown an impact of both long-term and short-term sources of cross-linguistic phonetic influence, it was anticipated that a shift in linguistic environment would result in a corresponding shift in phonetic production. Specifically, L1 English—L2 Spanish speakers would produce Spanish tokens with more English-like phonetic features in an English-dominant linguistic environment and more Spanish-like phonetic features in a Spanish-dominant environment.*

RQ2: Does proficiency in the L2 interact with linguistic environment?

**Hypothesis 2.** *Given the previous finding that as L2 experience and proficiency increase, L2 phonetic production shows less evidence of L1 to L2 cross-linguistic influence, it was anticipated that speakers with greater proficiency in the L2 may show smaller effects of a change in linguistic environment on their phonetic production.*

To assess the role of linguistic environment on phonetic production, the same participants were tested in both an English-dominant (Indiana, USA) and Spanish-dominant environment (Madrid, Spain). To focus squarely on the potential for linguistic environment to serve as a short-term source of cross-linguistic influence and limit the long-term effects of acquisition and immersion, participants were tested immediately prior to leaving one environment (i.e., less than 72 h pre-departure) and immediately upon arrival in the second environment (i.e., in the first 72 h in the new environment). The order of sessions was counterbalanced, such that one group was tested first in the English-dominant linguistic environment and then in the Spanish-dominant linguistic environment, and the other group received the opposite session order (i.e., Spanish-dominant environment then English-dominant linguistic environment).

Two different levels of phonetic analysis were conducted: an acoustic analysis of the voice onset time cue (VOT) and a global accent rating (GAR). These two measures were chosen to provide both a fine-grained measure of a relevant segment that differs cross-linguistically between English and Spanish (i.e., VOT), as well as a more global measure of perceived accent to capture potential shifts in other features (e.g., vowels, suprasegmental features, etc.) of production (i.e., GAR).

## 3. Methodology

### 3.1. Participants

Twenty English-speaking (L1) learners of Spanish (L2) participated in the current study. All participants were enrolled in a six-week immersive study abroad program in Madrid, Spain (Spanish-dominant environment) during the summers of either 2015 or 2018. The study abroad

program included host family stays, classes at the local university, and several day trips to surrounding cultural sites, all of which were conducted in the L2. Participants were all learners at the intermediate to advanced level, enrolled in 3rd or 4th year university language courses. All participants gave informed consent prior to beginning the task and the protocol was approved by the Institutional Review Board at Purdue University (Protocol #: 1303013396).

Immediately prior to the main task, participants completed the Bilingual Language Profile (BLP; Birdsong et al. 2012). The BLP relies on self-assessments of a participant's language history, language use, language proficiency, and language attitudes in each of the relevant languages to provide a composite language score. All participants are native speakers of English, having learned English from birth ($M$ age of acquisition = 0.0, $SD$ = 0.0) and Spanish after the age of 5 ($M$ = 12.1, $SD$ = 3.0). Across all components, participants self-rated English as higher than Spanish (Table 1).

**Table 1.** Unweighted results of the Bilingual Language Profile (BLP) subcomponents.

| Component | Scale [a] | English M (SD) | Spanish M (SD) |
|---|---|---|---|
| Language History | 0–120 | 104.2 (8.7) | 17.2 (8.4) |
| Language Use | 0–50 | 46.6 (2.8) | 3.3 (2.8) |
| Language Proficiency | 0–24 | 23.8 (0.6) | 14.2 (2.6) |
| Language Attitudes | 0–24 | 23.4 (2.8) | 15.0 (5.4) |

[a] For each scale, higher ratings correspond to a more engagement with that component of a given language.

Following the BLP scoring procedures (for details, see Birdsong et al. (2012)), a composite language score was computed for each participant in each language, giving equal weight to each subcomponent. The possible composite language score, henceforth referred to as proficiency, ranges from 0, corresponding to no proficiency in the language, to 218, indicating high proficiency in the language. As expected, participants reported high language scores for English ($M$ = 204.6, $SD$ = 7.9). In contrast, participants reported lower and more varied Spanish language scores ($M$ = 77.5, $SD$ = 18.4). Figure 1 illustrates the distribution of participants with respect to their Spanish language score. All participants are considered to be English-dominant. All participants reported normal speech and hearing.

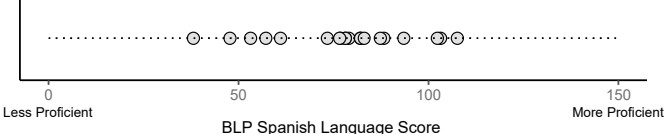

**Figure 1.** BLP Spanish language score by participant.

To assess the effect of linguistic environment, but counter-balance session order, participants ($N$ = 20) were divided into two mutually exclusive groups. The first group ($n$ = 12) was tested first in the English-dominant environment and then in the Spanish-dominant environment. The second group ($n$ = 8) was tested first in the Spanish-dominant environment and then in the English-dominant environment. Again, to limit any possible long-term sources of cross-linguistic influence, participants were tested immediately prior to departure from Environment A (< 72 h) and immediately upon arrival in Environment B.[2]

---

2   In short, one group of participants was tested en route to the host country of the study abroad program (i.e., prior to leaving Indiana, USA, and upon arrival in Madrid, Spain) and one group of participants was tested en route to the home country (i.e., prior to leaving Madrid, Spain, and upon arrival in Indiana, USA). While this presents a potential confound, such that one group may have had a six-week "advantage" by participating following six weeks of immersion in the host country, statistical analysis controlled for between-participant differences with the random effects structure, effectively comparing each participant to her or himself.

*3.2. Stimuli*

Stimuli for the read-aloud task were modified Spanish versions of utterances (*N* = 5) used in a global accent rating task (e.g., Flege 1988; Riney and Flege 1998). Relevant for the VOT analysis, embedded within these utterances were a number of tokens with word-initial voiceless stops. English and Spanish both employ a bi-partite distinction between voiceless and voiced stops in word initial position. VOT, defined as the temporal difference between the release of the oral closure and the onset of vocal fold vibration, has been shown to be a reliable cue to this voicing distinction (e.g., Lisker and Abramson 1964). Although both English and Spanish make use of this phonological distinction between voiceless and voiced phonemes, the phonetic cues differ. Specifically for voiceless stop consonants, English stops are produced with long-lag VOT (30–100 ms), while Spanish is produced with short-lag VOT (0–30 ms). Given this cross-linguistic difference, English-speaking learners of Spanish are tasked with acquiring and maintaining separate the Spanish-like short-lag VOT norms. A number of authors have noted that English-speaking learners of Spanish may produce Spanish voiceless stop consonants with English-like VOTs (e.g., Hammond 2001). Figure 2 shows the spectrogram and waveform for the word *<calle>* [kaje] 'street,' produced by a native English speaker (left) and native Spanish speaker (right). While the use of English-like VOT in Spanish voiceless stops is unlikely to cause issues of intelligibility (Lord 2005; Munro and Derwing 1995), it may impact the perception of speaker accentedness. Given both the gradient nature of VOT, and the previous evidence that bilinguals, including L2 learners, are able to effectively distinguish between English and Spanish VOT, VOT may serve as a sensitive measure of cross-linguistic influence between the two phonetic systems. A total of eight words contained word-initial voiceless stops, with the following distribution: /p/ = 2, /t/ = 2, /k/ = 4.

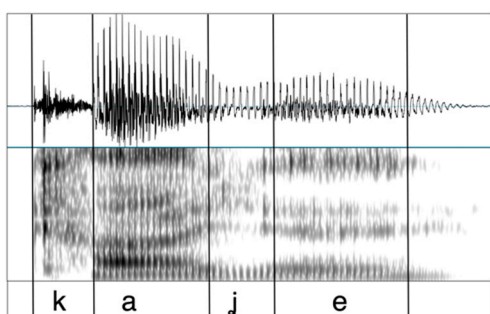 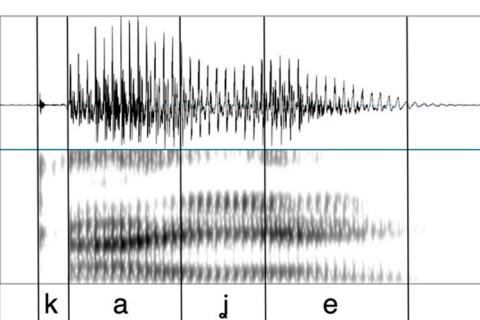

**Figure 2.** Spectrogram and waveform of *<calle>* 'street' produced by a native English speaker (**left**) and a native Spanish speaker (**right**). The difference in VOT is visible in the phoneme /k/.

Relevant for the GAR task, the five utterances contained a wide variety of segments (i.e., consonants and vowels). Several segments were included that differ significantly in English and Spanish phonetic production (e.g., word-initial <r>, which is produced as /ɹ/ in English and /r/ in Spanish), which could potentially serve as markers of English-accented Spanish. In addition, several segments were included that differ across different dialects of Spanish (e.g., <z>, <ci>, and <ce>, are produced as /s/ in most dialects of Spanish, but as /θ/ in Peninsular Spanish), which could potentially serve as markers of the local (Peninsular) dialect of Spanish.

Example (1) provides several sample stimuli. Target phonemes for the VOT analysis are underlined.

1.    a. T̲e voy a leer este p̲oema. 'I am going to read this poem to you.'
2.    b. ¿Doblaste a la derecha en la calle principal? 'Did you turn to the left on the main street?'
3.    c. Jaime c̲omió los c̲aramelos. 'Jaime ate the candies.'

### 3.3. Procedure

To assess the potential impact of linguistic environment, each speaker participated in two experimental sessions, each conducted in a different linguistic environment. One experimental session was conducted in an English-dominant linguistic environment (Indiana, USA), and one session was conducted in a Spanish-dominant linguistic environment (Madrid, Spain). In Indiana, USA, English is the home language of approximately 91.1% of the population, with Spanish spoken in the home by only 4.7% of the population (U.S. Census Bureau 2018). In Spain, 90% of the population speaks Spanish as a native language, while only 2.2% of the population speaks English as a native language (Instituto Nacional de Estadística 2019), although statistics were not available for the Community of Madrid. Moreover, both Indiana, USA, and Madrid, Spain, are considered to be "single language" environments in which little code-switching is present (Green and Abutalebi 2013). While the experience of each individual participant may vary, a finding captured by the language background questionnaire, the two environments are clearly distinguishable by the language of the broader environment.

Interaction with the experimenter was intentionally conducted in both languages in each session. The experimenter was a native speaker of midwestern American English who had spent several years living in Madrid, Spain and was proficient in the local Spanish dialect. Written instructions, provided before the start of the oral production task, were comprised of both English and Spanish. With the exception of the location of the two sessions, other experimental factors were maintained as equal as possible, using identical equipment, instructions, and consent forms and conducted with the same experimenter.

Following the collection of language background information, target utterances were presented visually using SuperLab v.5 (Cedrus Corporation 2015) and each utterance was repeated three times during each session. Utterances were recorded in a quiet room with a head-mounted microphone using Audacity v. 2.2.2 recording software. Both the instructions and the recording equipment were the same in each of the two different environments.

### 3.4. Voice Onset Time Analysis

A total of 960 tokens were considered in the initial VOT analysis (20 speakers × 8 tokens × 3 repetitions × 2 session = 960 tokens). Twenty-five tokens were classified as missing, and an additional 29 tokens were eliminated for a variety of speech errors (i.e., false start on target word) and recording errors (i.e., noisy recording). Lastly, outliers were eliminated ($n = 5$), defined as those tokens with VOT values greater than 3SD above and below the mean. A total of 901 tokens were included in the final VOT analysis.

VOT was defined as the temporal difference between the release of the oral closure and the onset of vibration of the vocal folds (e.g., Lisker and Abramson 1964). Tokens were measured using Praat (Boersma and Weenink 2018), with particular attention to the waveform. Tokens were coded blindly by a trained research assistant who was unaware of the linguistic environment in which the utterance was produced.

Statistical analysis was conducted using R statistical software (R Core Team 2013) and the lme4 package (Bates et al. 2015). Following recommendations by Barr et al. (2013), the maximal random effects structures that permitted model convergence were used. The significance criterion was set at $|t| > 2.00$. Power analysis was conducted with the simr package (Green and MacLeod 2016).

### 3.5. Global Accent Rating Analysis

Five native Spanish speakers from the target region (Madrid, Spain) were recruited as raters for the GAR task. All raters were native speakers of the target dialect, and using the BLP (Birdsong et al. 2012), were considered highly dominant in Spanish. The rating procedure was largely based on work by Riney and Flege (1998).

Two of the three repetitions per utterance were selected from each session for presentation to the native speakers. When possible, preference was given to the second and third repetitions of the stimuli. For productions containing speech errors, such as pauses or fillers, the first repetitions was substituted. Raters could listen to each presentation multiple times, if needed. The intensity of each utterance was scaled via script in Praat (Boersma and Weenink 2018) to 65 dB. The order of presentation was fully randomized. A total of 400 learner-produced utterances were selected for presentation to the native raters.

Raters were asked to provide accent ratings for each utterance on a 9-point Likert scale in which 1 corresponded to a "very strong non-native accent" and 9 a "native accent." Each accent rating was converted to a z-score on a by-rater basis to normalize for the different ranges of values used by each rater. A total of 2000 ratings were provided by native speakers (5 utterances × 2 repetitions × 2 sessions × 20 participants × 5 raters).

Statistical analysis was again conducted using R statistical software (R Core Team 2013) and the lme4 (Bates et al. 2015) and simr (Green and MacLeod 2016) packages. The maximal random effects structures that permitted model convergence were used (Barr et al. 2013).

## 4. Results

### 4.1. Voice Onset Time

An initial mixed effects model was conducted with VOT (ms) as the dependent variable and linguistic environment (English-dominant environment vs. Spanish-dominant environment) as the fixed effect. Random effects included participant and item (i.e., word), with random slopes and intercepts. Examination of a Q-Q plot confirmed that the residuals of the model were normally distributed. Contrary to the initial hypotheses, results from this initial model demonstrated no significant effect of linguistic environment on VOT, with similar VOTs produced in the English- ($M = 44.2$ ms, $SD = 19.4$ ms) and Spanish-dominant ($M = 43.6$ ms, $SD = 18.8$ ms) environments. The results for the fixed effects are available in Table 2 (for random effects and model equation, see Appendix A). Figure 3 shows the VOTs produced in each linguistic environment, separated by initial phoneme. Again, while we expect some differences in VOT across place of articulation (Cho and Ladefoged 1999), the key comparison is between VOT in the English and Spanish environments.

**Table 2.** Voice Onset Time (VOT) Model Fixed Effects.

| | Estimate | Std. Error | *t*-Value | Lower 95% | Upper 95% | *d* | 95% CI |
|---|---|---|---|---|---|---|---|
| Intercept (English Environment) | 44.45 | 3.37 | 13.175 | 37.70 | 51.19 | | |
| Spanish Environment | 0.22 | 1.85 | 0.119 | −3.47 | 3.91 | 0.03 | [−0.15, 0.22] |

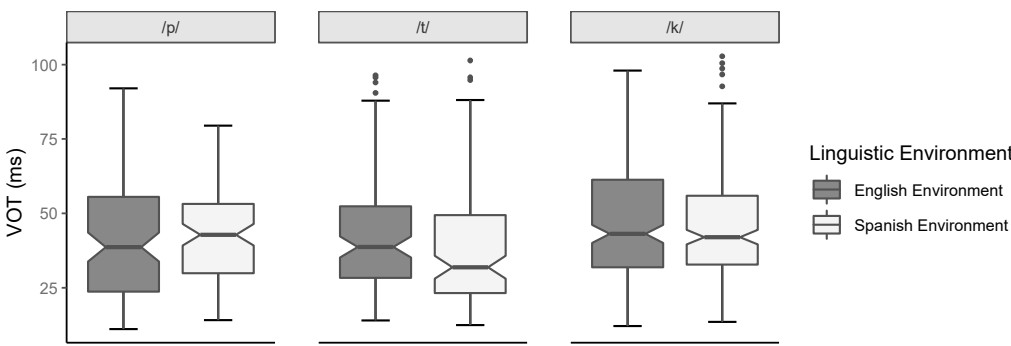

**Figure 3.** VOT (ms) by linguistic environment and place of articulation.

To ensure that the lack of a significant effect of linguistic environment on VOT production was not the result of an underpowered study, a power analysis was conducted using the simr package (Green and MacLeod 2016). Results of a simulation-based power analysis, with a medium effect size ($d = 0.5$) and based on 500 simulations, showed that the current study design surpassed the 80% power threshold (power for predictor linguistic environment = 99.8%, CI = [98.9, 100]). The outcome of the power analysis suggests that the lack of a significant effect of linguistic environment is not likely to be the result of an underpowered study design.

Related to the second research question, namely whether the effect of linguistic environment is conditioned by a given participant's proficiency in the target language, a second mixed effects model was conducted with the dependent variable of VOT. Fixed effects included linguistic environment (English-dominant environment vs. Spanish-dominant environment), proficiency, and their interaction. Proficiency was included as a continuous variable, with proficiency values determined by each participant's overall BLP language score for Spanish (Birdsong et al. 2012).[3] Random effects included participant and item, with random intercepts and random slopes by linguistic environment. More complex random effects structures, specifically random slopes by each of the two fixed effects, did not permit model convergence. Examination of a Q-Q plot confirmed that the residuals of the model were normally distributed. Results of this second model showed (see Table 3) that there was no significant effect of either linguistic environment or proficiency on VOT production. Moreover, there was no significant interaction between these two fixed effects, suggesting that linguistic environment was not a factor, regardless of a given participant's level of proficiency (for random effects and model equation, see Appendix B). Figure 4 illustrates these results. Again, lower VOT corresponds to more native-like pronunciation. Note that, while proficiency was included as a linear predictor in the model, Figure 4 grouped participants by relative proficiency for the purposes of visualization. Worth noting, the general expected trend is visible in Figure 4, with participants with higher proficiency in Spanish showing lower, more Spanish-like VOTs. However, it is clear that there is no effect of linguistic environment.[4]

**Table 3.** Voice Onset Time Model with Proficiency Fixed Effects.

|  | Estimate | Std. Error | *t*-Value | Lower 95% | Upper 95% |
|---|---|---|---|---|---|
| Intercept (English Environment) | 58.53 | 13.36 | 4.380 | 31.80 | 85.26 |
| Spanish Environment | 1.41 | 7.37 | 0.191 | −13.33 | 16.14 |
| Proficiency | −0.18 | 0.17 | −1.088 | −0.52 | 0.15 |
| Spanish Environment: Proficiency | −0.02 | −0.02 | −0.166 | 0.02 | −0.05 |

---

[3] Following suggestions by an anonymous reviewer, subsequent analysis was conducted with proficiency as a three-way categorical variable. Two different group cut-offs were considered. First, parallel to the subgroups in Figure 4, three approximately equal-sized proficiency groups were considered: low (n = 6), mid (n = 7), and high proficiency (n = 7). Second, three proficiency groups were identified using visual analysis of participant BLP Spanish score distributions: low (n = 5), mid (n = 12), and high proficiency (n = 3). Model comparison, following the procedure outlined below, showed that neither categorical approach to proficiency significantly improved model fit for either the VOT (equal sized groups: $\chi^2(4) = 1.380$, $p = 0.848$; unequal sized groups: $\chi^2(4) = 7.427$, $p = 0.115$) or the GAR analysis (equal sized groups: $\chi^2(4) = 3.517$, $p = 0.475$; unequal sized groups: $\chi^2(4) = 5.690$, $p = 0.224$) relative to a model without proficiency. As such, proficiency, regardless of operationalization, does not appear to significantly influence VOT or interaction with linguistic environment for this group of learners.

[4] As the main goal of this project was to examine the effect of linguistic environment, the main analysis compares the productions of participants in two different linguistic environments. The two groups (i.e., US-Spain and Spain-US) served to counter-balance session order. As such, some group differences are possible as the Spain-US group was tested following six weeks in the Spanish linguistic environment. Addressing the possible effect of group, a subsequent model was conducted with linguistic environment and group as fixed effects,. Model comparison showed that the inclusion of group significantly improved model fit ($\chi^2(2) = 7.395$, $p = 0.025$). This is not unexpected, given that overall, the Spain-US group ($M = 39.6$ ms, $SD = 18.3$ ms) produced significantly shorter VOTs than the US-Spain group ($M = 46.8$ ms, $SD = 19.1$ ms), $t(787) = 5.693$, $p < 0.001$. To confirm that the impact of linguistic environment was similar for each group, separate models were conducted for each group. The model structure was parallel to the main model above. Results suggested that linguistic environment did not significantly impact VOT for either the US-Spain group ($b = -2.616$, $t = -1.136$) or the Spain-US group ($b = 4.799$, $t = 1.923$).

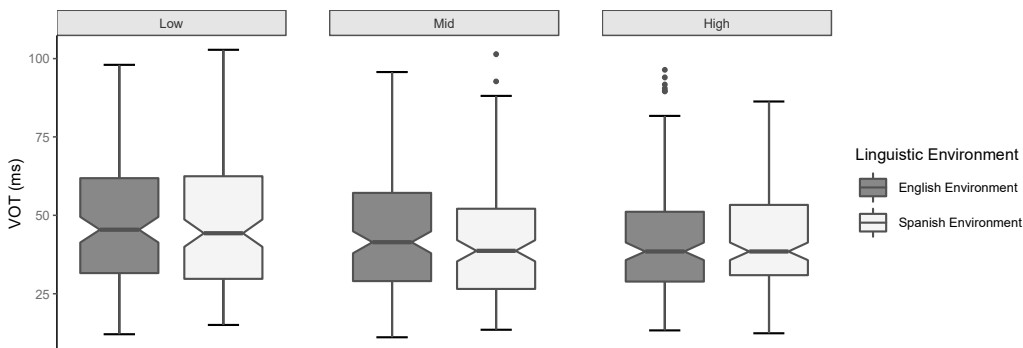

**Figure 4.** VOT (ms) by linguistic environment and proficiency. For the purposes of illustration only, participants were grouped into low (*n* = 6), mid (*n* = 7), and high proficiency (*n* = 7) groups by the continuous Spanish language component score of the BLP.

Finally, for completeness, a model comparison was done to assess the contribution of each of the variables of interest. Model comparison was conducted by comparing the model involving the two fixed effects (i.e., linguistic environment and proficiency), with the random effects structure previously detailed above, to submodels created by dropping one of the fixed effects but maintaining a similar random effects structure. Results of the model comparison show that there was no significant difference between the most complex model (log likelihood = −3706.4) and the submodel without the fixed effect of linguistic environment (log likelihood = −3706.4, $\chi^2$(2) = 0.041, p = 0.979). Similarly, there was no difference between the most complex model and the submodel without the fixed effect of proficiency (log likelihood = −3707.2, $\chi^2$(2) = 1.700, *p* = 0.428). As such, neither fixed effect contributed significantly to improving the model fit.

Taken as a whole, the results of the VOT analysis suggest that linguistic environment does not significantly impact the production of VOT. Moreover, for this group of learners, proficiency does not seem to be a relevant factor in the production of VOT.

*4.2. Global Accent Rating*

While the VOT analysis focuses on a specific segment, limited to only the voiceless stop consonants, the GAR analysis provides a more holistic metric of participant phonetic production. That is, while linguistic environment may not play a role in the production specifically of VOT, it is possible that other phonetic components, relevant to and noticeable by native speakers, may be modulated by environment.

Again, an initial mixed effects model was conducted with z-scored accent ratings as the dependent variable and linguistic environment as the fixed effect. Random effects included participant and item (i.e., utterance), with random slopes and intercepts. More complex random effects structures, particularly including rater as a random effect, did not permit model convergence. Again, each rating from the 9-point Likert scale was converted into a z-score on a by-rater basis. A visual analysis of the Q-Q plot confirmed that the residuals of the model were normally distributed. Results from this initial model closely parallel the results from the VOT analysis. Specifically, there was no significant impact of linguistic environment on accent ratings, with accent ratings for utterances produced in the English-dominant linguistic environment (*M* = -0.027, *SD* = 1.004) similar to those produced in the Spanish-dominant linguistic environment (*M* = 0.038, *SD* = 1.001). Full fixed effects results are seen in Table 4 (for random effects and model equation, see Appendix C).[5] Figure 5 illustrates the

---

[5] Parallel to the by-group analysis for VOT, mixed effect model was conducted on GAR with linguistic environment and group as fixed effects. Model comparison showed that the inclusion of group did not significantly improve model fit ($\chi^2$(2) = 5.143,

global accent ratings by linguistic environment. Again, a higher accent rating corresponds to a more native-like production.

**Table 4.** Global Accent Rating Model Fixed Effects.

| | Estimate | Std. Error | *t*-Value | Lower 95% | Upper 95% | *d* | 95% CI |
|---|---|---|---|---|---|---|---|
| Intercept (English Environment) | −0.027 | 0.147 | −0.185 | −0.321 | 0.267 | | |
| Spanish Environment | 0.065 | 0.082 | 0.795 | −0.099 | 0.229 | 0.06 | [−0.06, 0.19] |

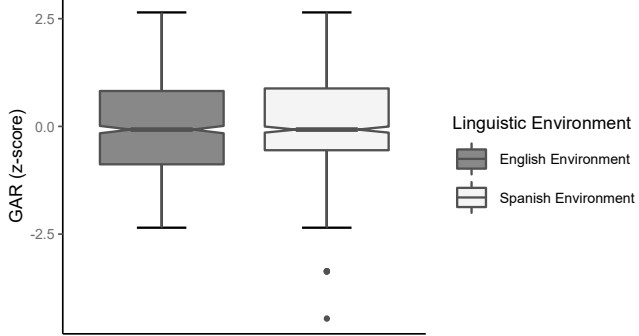

**Figure 5.** Global accent rating (z-scored) by linguistic environment.

To confirm that the lack of effect of linguistic environment on GAR was not due to an underpowered study, a power analysis was conducted using a simulation based-approach with the simr package (Green and MacLeod 2016). Results, based on 500 simulations with a medium effect size ($d = 0.5$), showed that the experiment exceeded the 80% threshold (power for predictor linguistic environment = 99.2%, CI = [97.9, 99.8]).

Considering the role of proficiency, a second model was conducted with linguistic environment and proficiency, as well as their interactions, as fixed effects. Participant and utterance were included as random effects, with random intercepts and slopes by linguistic environment, which was the maximal effects structure that permitted model convergence. Examination of a Q-Q plot confirmed that the residuals of the model were normally distributed. Results from the model with proficiency (for fixed effects, see Table 5; for random effects and model equation, see Appendix D) showed no significant effect of either linguistic environment or proficiency, and no significant interaction between the two fixed effects. Worth noting, the effect of proficiency trended in the expected direction, with participants who had higher self-rated Spanish language skills being rated as having more native-like accents.

**Table 5.** Global Accent Rating Model with Proficiency Fixed Effects.

| | Estimate | Std. Error | *t*-Value | Lower 95% | Upper 95% |
|---|---|---|---|---|---|
| Intercept (English Environment) | −1.013 | 0.608 | −1.666 | −2.229 | 0.203 |
| Spanish Environment | −0.006 | 0.287 | −0.022 | −0.580 | 0.568 |
| Proficiency | 0.012 | 0.008 | 1.667 | −0.003 | 0.027 |
| Spanish Environment: Proficiency | 0.001 | 0.004 | 0.259 | −0.006 | 0.008 |

Finally, a model comparison was conducted to examine the contribution of each of these fixed effects. The most complex model, with linguistic environment and proficiency as fixed effects, was compared to submodels created by dropping one of the fixed effects but maintaining the same random effects structure. Results from the model comparison showed that there was no significant

---

$p = 0.076$). As with VOT, results demonstrated that linguistic environment did not significantly impact the GAR for either the US-Spain group ($b = 0.13$, $t = 1.169$) or the Spain-US group ($b = −0.03$, $t = −0.572$).

difference between the complex model (log likelihood = −2407.7) and either the submodel without linguistic environment (log likelihood = −2408.0, $\chi^2(2) = 0.7496$, $p = 0.687$) or the submodel without proficiency (log likelihood = −2409.3.0, $\chi^2(2) = 3.315$, $p = 0.191$).

As with the analysis for VOT, the analysis from the GAR data suggests that linguistic environment did not impact native speaker ratings of learner productions. Moreover, while proficiency trended in the expected direction, it did not significantly interact with linguistic environment.

## 5. Discussion

The findings of this study add to the discussion on short-term sources of cross-linguistic influence and interaction. With respect to the first research question, and contrary to the initial hypothesis, the results showed that there was no significant effect of linguistic environment on cross-linguistic phonetic influence. The language of the broader community where an interaction took place had no relation to a bilingual's phonetic production. This result was found at both the fine-grained phonetic level, through an analysis of the VOT associated with Spanish voiceless stop consonants, as well as the more global level, as shown by the native speaker global accent ratings. Both VOT and perceived accent did not differ based on the linguistic environment of the session. Power analyses suggested that this lack of significant results is unlikely to be attributed to an underpowered study.

With respect to the second research question, namely the potential role of proficiency in modulating cross-linguistic phonetic influence, the results showed no significant role of proficiency for this particular group of speakers, and importantly, no interaction with the variable of linguistic environment. For all participants, regardless of proficiency, linguistic environment did not play a significant role in determining the degree of cross-linguistic influence or transfer. Again, this finding is contrary to the initial hypotheses.

The first hypothesis, specifically that linguistic environment would impact phonetic production, was driven by a robust body of research that has shown that bilingual phonetic production, and the degree of cross-linguistic phonetic interaction, is impacted by both long-term and short-term factors. Directly related to this hypothesis, we have seen that long-term immersion, either through immigration (Piske et al. 2001), travel (Sancier and Fowler 1997), or study abroad (Casillas 2020; Lord 2010; Solon and Long 2018), impacts phonetic production. Broadly, this research has shown that, over time, both a speaker's L2 and L1 (Bergmann et al. 2016; Major 1992; Stoehr et al. 2017) shift in the direction of the language of the broader community. Considering short-term sources of cross-linguistic phonetic influence, previous work has highlighted several short-term sources, including the language of a given interaction (e.g., Amengual 2018; Olson 2013; Simonet 2014), the use of code-switching (Antoniou et al. 2011; Balukas and Koops 2015; Bullock et al. 2006; Olson 2016), the presence of salient region-specific extra-linguistic cues in the interactional environment (Hay and Drager 2010), and visible (e.g., Babel and Russell 2015; Koops et al. 2008) and non-visible (Niedzielski 1999) social information about an interlocutor. Considering this line of research, it was anticipated that linguistic environment would impact production in the short-term, with phonetic targets shifting in the direction of the broader linguistic environment. Namely, it was anticipated that tokens produced in Madrid, Spain, would become more Spanish-like and tokens produced in Indiana, USA, would become more English-like. Given the lack of support for this hypothesis, it is worth considering several possible explanations.

One possible explanation for the lack of a short-term impact of linguistic environment is that the immediate local context of an interaction may be more relevant than the broader environment in which an interaction takes place. In much of the previous research on short-term sources of cross-linguistic phonetic influence, the source of the influence is present either in the interaction itself, either real (i.e., the language(s) required by the paradigm (e.g., Simonet 2014)) or imagined (i.e., visible or non-visible sociolinguistic cues (e.g., Babel and Russell 2015)), or is present in the physical environment that immediately surrounds participants (i.e., region-specific cues in the experimental setting (Hay and Drager 2010)). In each of these cases, the source of the short-term phonetic influence is in the speaker's immediate context. As such, the findings of the current study suggest a possible

primacy of this immediate context relative to the broader linguistic environment. In short, the immediate context of the interaction is more relevant for phonetic production and perception than the broader linguistic environment, and short-term sources of cross-linguistic phonetic influence are local, rather than global. In the current study, the immediate context was maintained as similar as possible across the two experimental sessions. The same experimenter greeted participants, and the language of both the interaction with the experimenter (i.e., bilingual/code-switched interaction) and the written instructions were the same in both sessions. If there exists a primacy of the immediate context for short-term sources of cross-linguistic phonetic influence, these local characteristics of the interaction may have been more relevant than the linguistic environment of the broader community.

A second possible explanation for the lack of an impact of linguistic environment on phonetic production in the current study is related to the study population. While previous research has shown that L2-learners' phonetic productions are impacted by both long-term and short-term sources of cross-linguistic phonetic influence, the participants in the current study have relatively low proficiency in the L2. Directly related to their phonetic systems, the mean VOT values produced by these participants (for all productions $M = 43.9$ ms, $SD = 19.1$ ms) remain well outside the norms for native speakers (e.g., Lisker and Abramson 1964) and both early and late bilinguals (Amengual 2012). As such, it is possible that these participants are not sufficiently proficient in the L2 to respond to short-term sources of cross-linguistic influence at the phonetic level. The possible role of proficiency as a mitigating factor is echoed in previous work on L2 development during long-term engagement with a given language (e.g., study abroad), in which lower-proficiency speakers evidence less change in the L2 than higher-proficiency speakers during the immersion experience (for discussion of a threshold hypothesis in study abroad, see Lafford and Collentine (2006)). As such, a speaker's proficiency level may serve to modulate the impacts of short-term sources and effectively limit the role of linguistic environment in the current population. Moreover, the population for this study was fairly homogenous at the phonetic level, as illustrated by the minimal differences in VOT between the highest and lowest proficiency groups (see Figure 4; mean difference = 6.1 ms). This degree of homogeneity may further explain the lack of an impact of proficiency and the failure to support the second hypothesis.

Finally, it is worth returning to the language mode framework to provide an account for the role of both long-term and short-term sources of cross-linguistic influence. Grosjean (2001) provides a variety of factors that influence language mode: the interlocutors, the situation and physical location, the function of the language act, the type of stimuli and task, etc. It is worth noting that all of these factors can be considered as short-term sources of cross-linguistic influence and are subject to change within and between different interactions. Grosjean (2001) notes that language mode "concerns the level of activation of two languages" (p. 42), and the short-term variables may be the primary drivers of shifts in language mode. In contrast, the long-term sources of linguistic interaction, including acquisition, changes in proficiency, and immersion, are not listed as factors that impact language mode. As language mode has been described as a continuum, from monolingual operation in Language A to monolingual operation in Language B, short-term factors may serve to adjust a participant's position along their existing continuum. Long-term factors, such as proficiency, may serve to manipulate the nature of the endpoints of this continuum. Additional support for this interpretation comes from work in bilingual lexical access, notably from picture-naming tasks. This line of work has shown that short-term factors, such as the ratio of one language to another, impacts lexical access in both production (e.g., Gollan and Ferreira 2009; Olson 2015) and perception (e.g., Olson 2017), but that these effects are modulated by more long-term oriented factors like proficiency and language dominance (e.g., Olson 2015; Schwieter and Sunderman 2008). In terms of activation, long-term factors may ultimately manipulate a given language's baseline activation level or range of possible activation, while short-term factors manipulate the comparative level of activations of the two languages within their possible ranges.

*Future Directions*

Future research should continue to systematically examine the differential impacts of both long- and short-term sources of cross-linguistic phonetic influence. Building directly on the current study, and particularly in light of the failure of the results to support the initial hypotheses, future research may seek to expand upon the current study with a more heterogeneous population. Of particular interest may be to examine participants across a wide range of proficiencies, from learners to highly proficient early bilinguals, as they move through different linguistic environments. Moreover, developing a better understanding of other individual factors, such as participant's engagement with the local context (i.e., whether they maintain significant use of the L1 or immediately begin to engage in the L2), may serve to further our understanding of variability in cross-linguistic phonetic interaction. Second, it is acknowledged that the data set for the acoustic analysis is limited, both in terms of the number of tokens and the variety of features examined. Future research may seek to replicate these findings with a larger data set and a variety of phonetic features. For example, a more robust analysis across different places of articulation, precluded here by the size of the data set, may also be of interest. Notably, there appears to be some slight advantage for /t/ relative to the other places of articulation (see Figure 3), which would suggest that different phonemes may undergo different levels of cross-linguistic influence (for a discussion of potential differences in the perceptual prominence of voiceless stops by place of articulation, see Ruch and Peters (2016, p. 28)). Furthermore, research may seek to disentangle the possible effects of the immediate interactional context and the broader environment of that context, exploring the possible notion of a primacy of immediate or local factors as short-term sources of cross-linguistic influence.

## 6. Conclusions

This study examined the potential for linguistic environment, conceptualized as the language norms of the broader community in which an interaction or experiment takes place, to serve as a short-term source of cross-linguistic influence. To assess the role of linguistic environment, bilinguals (i.e., English speaking learners of Spanish) produced Spanish utterances in two sessions: an English-dominant linguistic environment (Indiana, USA) and a Spanish-dominant linguistic environment (Madrid, Spain). Productions were analyzed at the fine-grained acoustic level, though an acoustic analysis of voice onset time, as well as more holistically through native speaker global accent ratings. Results showed that linguistic environment did not significantly impact either measure of phonetic production. Moreover, there was no interaction of proficiency with linguistic environment, suggesting that the linguistic environment was not a relevant factor, regardless of participant proficiency in their second language.

The current findings, notably the lack of an impact of the broader linguistic environment in determining cross-linguistic phonetic influence, may suggest a primacy of the local factors (i.e., characteristics of the interaction and the immediately surrounding area) over broader, global factors (i.e., the linguistic environment of the broader community surrounding the interaction) as sources of short-term cross-linguistic interaction. Further research is needed to confirm these results and continue to explore the role of both long-term and short-term sources of cross-linguistic phonetic influence.

**Funding:** This research was funded in part by the Purdue Research Foundation.

**Acknowledgments:** I am grateful for the technical work of Serae Neidigh on this project. All errors are my own.

**Conflicts of Interest:** The author declares no conflict of interest. The funding sponsors had no role in the design of the study; in the data collection, analyses, or interpretation of the data; in the writing of the manuscript, or in the decision to publish the results.

## Appendix A

**Table A1.** Voice Onset Time Model Random Effects.

| Participant | Variance | Std. Dev. | Corr. |
|---|---|---|---|
| Intercept | 172.98 | 13.15 | |
| Spanish Environment | 34.33 | 5.59 | −0.36 |
| **Item** | **Variance** | **Std. Dev.** | **Corr.** |
| Intercept | 15.95 | 3.99 | |
| Spanish Environment | 5.36 | 2.31 | 0.09 |

Equation:

$$VOT_{ijk} = \beta_0 + \beta_1 * I\left(Environment_{ijk} = B\right) + Participant_j + Item_k + \epsilon_{ijk}$$

## Appendix B

**Table A2.** Voice Onset Time Model with Proficiency Random Effects.

| Participant | Variance | Std. Dev. | Corr. |
|---|---|---|---|
| Intercept | 171.17 | 13.08 | |
| Spanish Environment | 37.17 | 6.10 | −0.39 |
| **Item** | **Variance** | **Std. Dev.** | **Corr.** |
| Intercept | 15.95 | 3.99 | |
| Spanish Environment | 5.37 | 2.32 | 0.09 |

Equation:

$$\begin{aligned} VOT_{ijk} = {}& \beta_0 + \beta_1 * I\left(Environment_{ijk} = B\right) + \beta_2 * Proficiency + \beta_3 \\ & * I\left(Environment_{ijk} = B\right) * Proficiency + \gamma_j * I(Environment_{ijk} \\ & = B) + \eta_k * I\left(Environment_{ijk} = B\right) + Participant_j + Item_k + \epsilon_{ijk} \end{aligned}$$

## Appendix C

**Table A3.** Global Accent Rating Model Random Effects.

| Participant | Variance | Std. Dev. | Corr. |
|---|---|---|---|
| Intercept | 0.399 | 0.632 | |
| Spanish Environment | 0.053 | 0.230 | −0.17 |
| **Item** | **Variance** | **Std. Dev.** | **Corr.** |
| Intercept | 0.005 | 0.072 | |
| Spanish Environment | 0.014 | 0.119 | −0.2 |

Equation:

$$GAR_{ijk} = \beta_0 + \beta_1 * I\left(Environment_{ijk} = B\right) + Participant_j + Item_k + \epsilon_{ijk}$$

## Appendix D

**Table A4.** Global Accent Rating Model with Proficiency Random Effects.

| Participant | Variance | Std. Dev. | Corr. |
|---|---|---|---|
| Intercept | 0.364 | 0.604 | |
| Spanish Environment | 0.056 | 0.238 | −0.22 |
| **Item** | **Variance** | **Std. Dev.** | **Corr.** |
| Intercept | 0.005 | 0.072 | |
| Spanish Environment | 0.014 | 0.119 | −0.2 |

Equation:

$$
\begin{aligned}
GAR_{ijk} = {} & \beta_0 + \beta_1 * I\big(Environment_{ijk} = B\big) + \beta_2 * Proficiency + \beta_3 \\
& * I\big(Environment_{ijk} = B\big) * Proficiency + \gamma_j * I\big(Environment_{ijk} \\
& = B\big) + \eta_k * I\big(Environment_{ijk} = B\big) + Participant_j + Item_k + \epsilon_{ijk}
\end{aligned}
$$

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
