# Peer review of "Short-Term Sources of Cross-Linguistic Phonetic Influence: Examining the Role of Linguistic Environment"

_languages, doi:10.3390/languages5040043_

Round 1

Reviewer 1 Report

Overall Evaluation

I thank the author(s) for a very interesting and timely read. I agree with the author that the role of language environment as a short-term source of cross-linguistic interaction has been understudied. In fact, I believe the short-term source of cross-linguistic interaction (whether language mode, code-switching, etc.) is more understudied than the long-term sources of cross-linguistic interaction, which makes this article a very welcome addition. The paper presents a very solid experimental design, statistical analyses, and data visualization. Additionally, the writing style of the author is excellent. This is always a pleasure to read something that flows so well. It was very enjoyable reading this article and I thank the author(s) for their attention to detail. The literature review was quite good in providing needed information. While overall, I find this study quite interesting with the ability to inspire more work on the effect (or lack thereof) of language environment, there are a few points that need to be addressed. Here I list my major concerns and below they are detailed further in the detailed comments section.

Complex models: The analysis itself is quite sound, the mixed effects linear regression models with both random intercepts and slopes are superbly. However, I wonder if given relatively few participants as well as tokens, if random slopes are needed. See my comments below, but I think the random intercepts (word and participant) are the only items needed so as not to make these models too complicated that may result in a Type II error.

Proficiency: The notion of Proficiency needs to be explored more as this is crucial to the findings and I think could better connect this study not only to short-term sources of cross-linguistic interaction, but also to a growing body of study abroad literature. It would be helpful to have more information about these L2 students to know if they are all in the same “proficiency level” deemed by the institution (i.e. advanced, intermediate). Several study abroad studies determine proficiency (at start of program) by grammar tests or simple the course level of the students. It would be helpful to know this. Also, regarding their proficiency, is there any way to know that these speakers have acquired the short-lag / long-lag VOT language-specific contrast in order to be able to show a language environment effect? I think addressing this would make the paper stronger and better defend possible critiques. As I mention below, I feel that proficiency might be better seen here in three-categories: low, mid, high. While I think continuous IVs are always better than categorical IVs, I wonder if it actually would be more beneficial here to look at proficiency as determined by the BLP as a three-way categorical variable. Given there are only 20 participants, this might be difficult for significance for a continuous variable. Additionally, could the author find a way to defend the notion that those who were recorded at the end of a study abroad rated themselves higher than those who were recorded at the beginning of the program? See below for more comments.

Place of articulation: I would recommend included place of articulation as an IV in the models. Figure 1 looks quite promising for some type of effect for the dental /t/ environment. Hanna Ruch and colleagues, examining post-consonant aspiration in Andalusian Spanish have proposed that the /th/ is more salient than the /ph/ or /kh/ environments. Thus, perhaps /t/ is the environment most effected (or maybe the only VOT environment that these L2 speakers are capable of separating in the first place, thus a place where environment could have an impact).

Interactions: I would highly recommend exhausting all interactions: two and three-way interactions. That is, a three-way interaction between Environment X Proficiency (three-way categorical IV) X Place of articulation (bilabial, dental, velar) may yield some type of significant effect. That is, perhaps the mid or advanced groups show the effect of environment, but only for /t/, which would be an interesting result.

After a detailed review, I recommend this manuscript for a Accept with majors revisions/ Revise & Resubmit. Many of the comments below are stylistic in nature, so I leave many of these up to the discretion of the author. I hope the suggestions given help the author to make an already strong paper that much stronger. Again, I wanted to emphasize what a pleasure it was to read this article. As I mentioned above, the statistical practices, experimental design, and data visualization were very impressive. Best of luck in the revisions.

Specific comments (section by section):

ABSTRACT

Line 16: “though” à “through”

Line24: delete final coma at the end of the keywords

1 INTRODUCTION

Line 36: I recognize the author later defines what constitutes “long-term” versus “short-term” in more depth in the literature, but it might help the reader sooner than later to have a very brief distinction. As I was reading it I wondered where something like a 4-6 week study abroad would fall (which I believe is what the students from Indiana, USA partook in here). I think this would in fact be considered “long-term” in the current article, but the study abroad literature might consider this short-term (at least “short-term” study abroad). All this to say, perhaps even a short one sentence to set up where the division between the distinction lies.

Line 69: In addition to Casillas (2020) see also Solon & Long (2018).

Solon, M., & Long, A.Y. (2018). Acquisition of phonetics and phonology abroad: What we know and how. In Sanz, C., & Morales-Front, A. (Eds.), The Routledge Handbook of Study Abroad: Research and Practice (pp. 71-85). Routledge.

p.2 – Footnote1. I think variable should be “variables”, no?

2 LITERATURE REVIEW

Lines 101-102: In the study abroad literature, there is generally a separation between a short-term (4-6 weeks) and a long-term (1 semester or academic year) immersion. Thus, it might be good to mention from weeks (citations) to months (citations), to years (citations). I leave this at the discretion of the author(s) as I imagine there are some word limitations to the article that I haven’t been privileged to. In general, findings for a short-term study abroad aren’t great from language-specific acquisition (read: changes/gains). Lord (2010) found improvement in /bdg/ with impressionistic analysis. Bongiovanni et al. (2015) didn’t find so many gains in a short-term study. However, short-term study abroad in Argentina appears to demonstrate large gains in acquisition of regional features such as [ʃ]/[Ê’] as seen in Schmidt (in press). Long term study abroad in Argentina (see Pozzi & Bayley, 2020) appear to acquire the regional norm [ʃ]/[Ê’]. Raish (2015) also found the acquisition of the Egyptian regional [g] among some students in a long-term study abroad. All this to say, it might be good in mention that even weeks (as opposed to months and years) can have an impact in phonetic production.

Bongiovanni, S., Long, A., Megan S., & Willis, E.W. (2015). The effect of short-term study abroad on second language Spanish phonetic development. Studies in Hispanic and Lusophone Linguistics, 8(2), 243-283.

Pozzi, R. & Bayley, R. (2020). The development of a regional phonological feature during a semester abroad in Argentina. Studies in Second Language Acquisition. First view.

Raish, M. (2015). The acquisition of an Egyptian phonological variant by U.S. students in Cairo. Foreign Language Annals, 48(2), 267-283.

Schmidt, L.B. (forthcoming). Role of development language attitudes in study abroad context n adoption of dialectal pronunciations. Foreign Language Annals.

p.3 Footnote 2: It might be nice to have a page number here.

p.4 Perhaps Casillas & Simonet (2018) would be worth adding to the section that bilinguals maintain two systems, in production and perception in the case of their study.

Casillas, J.V., & Simonet, M. (2018). Perceptual categorization and bilingual language modes: Assessing the double phonemic boundary in early and late bilinguals. Journal of Phonetics 71, 51-64.

p.5 Line188 – See also Koops et al. (2008) study where images of young, middle age, and older females were placed a on a screen and they observed how visuals affected the perception of the pin-pen merger in Houston in a forced-choice task. It’s a fascinating study of how the social knowledge of age affected what they heard (given there is a change in progress in Houston in which the pin-pen merger is splitting among younger Houstonians).

Koops, Christian, Gentry, Elizabeth, & Pantos, Andrew (2008). The effect of perceived speaker age on the perception of PIN and PEN vowels in Houston, Texas. University of Pennsylvania Working Papers in Linguistics 14(2):93-101.

p.5 lines219-221 – This is a very helpful distinction here between short-term and long-term. Perhaps place copy this and place it in the introduction just to let readers know desde el principio what this distinction means.

p.6 line258: Does the global accent rating (GAR) have a citation? I’m unfamiliar with the accent rating literature, so forgive my ignorance, but is there a citation needed here? If not, no worries. As the author mentions it’s a Likert scale, perhaps its general enough that it doesn’t need one.

The author should make sure to include the work of D. Olson regarding short-term sources of cross-linguistic influence. To name a few (there are several that one should cite):

Olson, D.(2013). Bilingual language switching and selection at the phonetic level: Asymmetrical transfer in VOT production. Journal of Phonetics 41: 407-420.

Olson, D. (2016). The role of code-switching and language context in bilingual phonetic transfer. Journal of International Phonetic Association 46(3): 263-285.

3 METHODOLOGY

p.6 line 271: In regards to the BLP, did they all take the BLP at the same time (all at the beginning of the short-term study-abroad) or prior to their task (half at the beginning of the study abroad- half at the end of the study abroad)? If it was staggered, this could affect the ratings of proficiency. That is, students after 6 weeks may have rated themselves more proficient than they would have 6 weeks prior.

p.7 lines286-292: Perhaps I missed it, but it might be nice to mention the year these recordings took place.

p.7 line 312: This seems like a small amount of tokens per place of articulation.

p.7 Footnote 3: The font size appears larger. More importantly, it appears one group had a 6-week advantage over the other in acquiring more Spanish-like VOT. Although of note, which could be a way to defend this is Díaz-Campos and Lazar (2003) acoustically analyzed /ptk/ in pre and post in a semester long program and there were no gains for the SA group. This might help any critiques about the 6-week advantage. See also Bongiovanni et al. (2015), there were some subtle gains in a short-term SA for /ptk/, but no different than the at home students.

Díaz-Campos, M., & Lazar, N. (2003). Acoustic analysis of voiceless initial stops in the speech of study abroad and regular class students: Context of learning as a variable in Spanish second language acquisition. In P. Kempchinsky & C.E. Piñeros (Eds.), Theory practice, and acquisition (pp.352-370). Somerville, MA: Cascadilla.

p.8 line 317. It might be worth mentioning that it isn’t only the <z>, but also the <ci> and <ce> contexts where the interdental fricative is produced with distinción.

p.8 line 337 – I see a citation from 2001 with the Social Science Data Analysis Network. I think it might be better to have a more updated percentage of Spanish spoken at home. I would recommend using the American Community Survey: https://data.census.gov/cedsci/

In theory one should be able to have data for the year 2018 (click on year, geography, and language) in order to get the desired information. I can’t imagine the percentage is much higher in 2018 (although the Midwest has had much more immigration of Spanish-speakers in recent years), but I think it would look more accurate and up to date than 2001. The American Community Survey is the estimations per year based on the U.S. Census. I don’t believe the 2020 U.S. Census data will be available for quite some time still.

p.8 line 343: Was there any questionnaire about their Spanish-use outside the classroom?

p.8 line 354: This seems like a relatively low number of tokens for statistical power. However, it looks like the author(s) did a very sufficient job of exploring whether or not this was true.

p.8 line360 – If space allows, I think it’s always best to have an example of an ideal short-lag and a long-lag Praat Picture from the tokens, in case this article is read by non-phonetics experts. Or perhaps at least one picture with the VOT section segmented.

p.8 lines 262-263: I imagine these were research assistants, no?

p.9 Line 381: 2,000 with coma

4 RESULTS

p.9-10: Here I detail several things to consider in the statistical analysis:

RANDOM SLOPES: The author(s) do a great job here in making sure to avoid any Type I errors through a combination of random effects and slopes. While there is a great deal of debate on how complicated to make the models themselves, I for one am not convinced that more complicated is necessarily better. That is, I think as long as participant and word are incorporated as random effects, the random slopes might not be needed. Again, this is completely my position on the matter and respect if the author disagrees. I say this only to suggest trying these models without the slopes and see if anything significant results. If not, no worries, and continue with the more complicated models.

PROFICIENCY: While I think continuous IVs are always better than categorical IVs, I wonder if it actually would be more beneficial here to look at proficiency as determined by the BLP as a three-way categorical variable. Given there are only 20 participants, this might be difficult for significance for a continuous variable. That is, I feel like a low number of participants might be better situated in categorical proficiency levels, as opposed to a continuous one. What proficiency are these students? Is this an upper-level study abroad? An intermediate-level study abroad? Figure 2 calls my attention in which those labeled low show relatively high VOT with no difference between environment, high show relatively low VOT with no difference between environment, but then the mid proficiency group shows somewhat of a drop in the Spanish environment. This appears to be something of interest, that a continuous measure would not show. Perhaps it’s only the mid-proficiency that is affected by environment, with the low-proficiency always long-lag VOT and the high-proficiency always short-lag proficiency, regardless of environment. Thus, one could plot a BLP Dominance Score boxplot, similar to Amengual (2018 Figure 1) and see if there is a decent cut off point between groups and divide them as such- or some type of justification along these lines.

PLACE OF ARTICULATION: I wonder if the author included place of articulation as a fixed factor if there might be a significant result for bilabials, dentals, or velars. Figure 1 looks quite promising for some type of effect for the dental /t/ environment. Hanna Ruch and colleagues, examining post-consonant aspiration in Andalusian Spanish have proposed that the /th/ is more salient than the /ph/ or /kh/ environments. Thus, perhaps this is the environment most effected (or maybe the only VOT environment that these L2 speakers are capable of separating in the first place, thus a place where environment could have an impact).

INTERACTIONS: I would experiment with more interactions. That is, a three-way interaction between Environment X Proficiency (three-way categorical IV) X Place of articulation (bilabial, dental, velar) may yield some type of significant effect. That is, perhaps the mid or advanced groups show the effect of environment, but only for /t/.

p.11 lines 441-443: Regarding proficiency, did the author(s) test these L2 participants in English and contrast their VOT for /ptk/ between the two languages to see if they had the contrast to start with? The boxplots in Figure 2 indicate mostly long-lag VOT values above 30ms.

p.11 Footnote 5: This makes sense with so few tokens. Incredibly complex mixed effects models need a ton of data. Thus, perhaps too complex of models might actually produce a Type II error in this case.

p.12 Table4: Similar to the VOT, I would recommend trying three-way interactions with Environment by Proficiency (three-way categorical IV) X Group (those who were recording at the beginning of SA and those who were recorded at the end of SA).

p.12 Table 5: Given environment:proficiency is tested here, I don’t see the need to have the other table above it with only the same effect tested for proficiency. That is, it’s tested here in Table 5 so need for Table 4.

4 DISCUSSION

p.13 lines 498-499: This is a very helpful to defend the relatively low token count. Nice.

p.13 line 510: Perhaps add a few of the study abroad articles mentioned above here as well.

p.13 line 518: See also Walker et al. (2014) for an interesting matched-guise study in which the social information impacted how listeners heard /s/ aspiration.

Walker, Abby, García, Christina, Cortés, Yomi, & Campbell-Kibler, Kathryn (2014). Comparing social meanings across listener and speaker groups: The indexical field of Spanish /s/. Language Variation and Change 26(2), 169-189.

p.13 line 526: Should “immediately” say “immediate”?

p.14:  I think the role of proficiency should be emphasized much more than it is, even from the introduction and background. This is something that is understudied in the study abroad context. See Issa & Zalbidea (2018) for a very helpful review on the role of proficiency in study abroad gains, or lack thereof. Lafford & Collentine (2006) propose that cognitive abilities of more proficiency students are able to focus on linguistic variation in addition to communication, while the cognitive limitations of less proficient students only allows them to focus on communication (and not notice linguistic variation). I think proficiency could be better examined with a categorical IV (low, mid, high) which would better address the issue of whether or not environment varies per proficiency level.

Issa, B., & Zalbidea, J. (2018). Proficiency levels in study abroad: Is there an optimal time for sojourning? In C. Sanz & A. Morales-Front (Eds.), The Routledge Handbook of Study Abroad Research and Practice (pp.453-463). New York: Routledge.

Lafford, B., & Collentine, J. (2006). The effects of study abroad and classroom contexts on the acquisition of Spanish as a second language. In R. Salaberry & B. Lafford (Eds.), The Art Of Teaching Spanish: Second Language Acquisition: From Research To Praxis (pp. 103-126). Washington D.C: Georgetown University Press.

Reviewer 2 Report

The paper is very welcome as it enlarges the knowledge about factors affecting L2 phonetic attuning in a framework close to Flege’s SLM. The role played by the linguistic environment is explored with appropriate techniques and relevant results in the light of the more recent literature on bilingualism and phonetic interaction.

A too small number of tokens is measured in some cases and the statistical analysis could be improved in this direction.

Generally speaking, the effects of the linguistic environment may be judged of weak influence as emerges from the dispersion of the VOT values and boxplots. The only partial exception is offered by the realisations of /t/, thus reinforcing the idea of a different status of this phoneme in phonetic terms as compared with other aspirated stops.

Conclusions may be slightly improved by including similar considerations.

Author Response

REVIEWER 2
The paper is very welcome as it enlarges the knowledge about factors affecting L2 phonetic attuning in a framework close to Flege’s SLM. The role played by the linguistic environment is explored with appropriate techniques and relevant results in the light of the more recent literature on bilingualism and phonetic interaction.

Thank you.

A too small number of tokens is measured in some cases and the statistical analysis could be improved in this direction.

This is certainly a limitation, which we have acknowledged in the manuscript.

Lines 640-643: Second, it is acknowledged that the data set for the acoustic analysis is limited, both in terms of the number of tokens and the variety of features examined. Future research may seek to replicate these findings with a larger data set and a variety of phonetic features.

However, this is something that we think is mitigated in several specific ways. First, the use of the mixed effect model structure, namely with token as a random effect, allowed for direct comparison between items.  Second, use of a simulation-based power analysis suggested that the results are unlikely to be significantly impacted by the size of the data set.

Generally speaking, the effects of the linguistic environment may be judged of weak influence as emerges from the dispersion of the VOT values and boxplots. The only partial exception is offered by the realisations of /t/, thus reinforcing the idea of a different status of this phoneme in phonetic terms as compared with other aspirated stops. Conclusions may be slightly improved by including similar considerations.

I’ve added some discussion of this, notably in terms of what it means for cross-linguistic phonetic influence, in the discussion section.

642-647: Future research may seek to replicate these findings with a larger data set and a variety of phonetic features. For example, a more robust analysis across different places of articulation, precluded here by the size of the data set, may also be of interest. Notably, there appears to be some slight advantage for /t/ relative to the other places of articulation (see Figure 3), which would suggest that different phonemes may undergo different levels of cross-linguistic influence (for notions of perceptual prominence and the status of /t/ in Spanish see Ruch and Peters 2016).

Reviewer 3 Report

The present study examines the effects of short-term language immersion and language proficiency in the production of VOT and Spanish and English accents by native English learners of Spanish.  The results show no effects of short-term language experience or proficiency in both local (VOT) and global (overall accent quality) variables. The author(s) suggest this lack of effects could be due to the interference of other variables, such as the type of language interactions during the experiment.

The manuscript is very well written and structured, and the topic falls under the scope of the journal. The research questions are well motivated and the logistics involved in the abroad testing is definitively a big plus.

I believe the manuscript has the potential to be published in this journal. There are, however, a few questions that I would recommend to address in a new revision round:

Introduction

[1] This section is quite clear and complete. As a minor suggestion, the author(s) may consider adding the surrounding auditory context as another potential source of cross-linguistic interaction: Gonzales, K., & Lotto, A. J. (2013). A bafri, un pafri: Bilinguals’ pseudoword identifications support language-specific phonetic systems. Psychological science24(11), 2135-2142.

Methods / Results

[2] The authors place the description of the linear mixed-effects models in the Results. I feel these model descriptions are better suited in the Methods.

[3] Please add the equations of the linear mixed-effects models in addition to the verbal descriptions.

[4] The study includes two groups of participants. In the first group, participants are tested in Indiana a few days before traveling to Spain and then tested again in Spain a few days after their arrival. In the second group, participants are tested in Spain a few days before flying back to Indiana and then tested again in Indiana a few days after their arrival. I wonder why this important experimental feature is not included in the linear mixed-effects models that examine the effects of short-term immersion. The amount of immersion in the Spanish norm is quite different between groups, as they are tested at different times after their arrival in Spain. Does this group difference play any role in the results?

[5] None of the linear mixed-effects models seem to include interaction effects; or at least, these effects are not explicitly reported in the manuscript. I wonder why, as I find that the interactions between group and short-term immersion, and between short-term immersion and proficiency, are quite relevant for the aims of the study. Did the interactions worsen the model fit?  

[6] One or two more figures showing the results split by group would be extremely useful.

[7] The authors mention that the experimenters mixed both language norms. Could you be more specific about it? Were the experimenters instructed to mix the norms? Were they (the experimenters) native speakers of English, Spanish, early bilinguals, late bilinguals?

[8] Also: did you provide any written instructions during the experiment? If “yes” please indicate in which language.

[9] Do the experimental groups share any participants?

[10] In lines 414-415, the author(s) note: "more complex random effect structures did not permit model convergence". Which specific more complex effects were added to these models?

Discussion

[11] In my personal experience, the degree of engagement with a foreign language community in study abroad programs is quite variable across students. While some students love to engage since the very first day, others are much more willing to interact with their native language peers. As a minor suggestion, the author(s) might also consider discussing the potential effect of this in the results.

Round 2

Reviewer 3 Report

The authors have addressed my comments quite appropriately. I respect their decisions for the location of model descriptions and the requested (and finally excluded) figures. I have no additional comments.

Author Response

While the reviewer did not provide any additional comments, I upload here the response to comments from the other reviewer (Reviewer 1).